# An Adversarial Generative Network Designed for High-Resolution Monocular Depth Estimation from 2D HiRISE Images of Mars

**Riccardo La Grassa** [1,*,†] , **Ignazio Gallo** [2,†] , **Cristina Re** [1,†] , **Gabriele Cremonese** [1,†] , **Nicola Landro** [2,†] , **Claudio Pernechele** [1,†] , **Emanuele Simioni** [1,†] **and Mattia Gatti** [2,†]

1   INAF-Astronomical Observatory Padua, 35100 Padua, Italy
2   Department of Theoretical and Applied Science, University of Insubria, 21100 Varese, Italy
*   Correspondence: riccardo.lagrassa@inaf.it
†   These authors contributed equally to this work.

**Abstract:** In computer vision, stereoscopy allows the three-dimensional reconstruction of a scene using two 2D images taken from two slightly different points of view, to extract spatial information on the depth of the scene in the form of a map of disparities. In stereophotogrammetry, the disparity map is essential in extracting the digital terrain model (DTM) and thus obtaining a 3D spatial mapping, which is necessary for a better analysis of planetary surfaces. However, the entire reconstruction process performed with the stereo-matching algorithm can be time consuming and can generate many artifacts. Coupled with the lack of adequate stereo coverage, it can pose a significant obstacle to 3D planetary mapping. Recently, many deep learning architectures have been proposed for monocular depth estimation, which aspires to predict the third dimension given a single 2D image, with considerable advantages thanks to the simplification of the reconstruction problem, leading to a significant increase in interest in deep models for the generation of super-resolution images and DTM estimation. In this paper, we combine these last two concepts into a single end-to-end model and introduce a new generative adversarial network solution that estimates the DTM at 4× resolution from a single monocular image, called SRDiNet (super-resolution depth image network). Furthermore, we introduce a sub-network able to apply a refinement using interpolated input images to better enhance the fine details of the final product, and we demonstrate the effectiveness of its benefits through three different versions of the proposal: SRDiNet with GAN approach, SRDiNet without adversarial network, and SRDiNet without the refinement learned network plus GAN approach. The results of Oxia Planum (the landing site of the European Space Agency's Rosalind Franklin ExoMars rover 2023) are reported, applying the best model along all Oxia Planum tiles and releasing a 3D product enhanced by 4×.

**Keywords:** super resolution; 3D mapping; digital terrain model; deep learning; remote sensing; satellite images; Mars

## 1. Introduction

Stereoscopic vision is an essential technique for obtaining 3D information from 2D images. One of the most difficult problems for a stereo vision system is the problem of stereo matching. This is the process of recovering depth from a set of two-dimensional overlapping images taken from different positions covering the same scenario [1]. Traditional methods of image-based depth estimation usually rely on a binocular camera to calculate the disparity of two 2D images (taken by a binocular camera) by stereo matching. Once the corresponding points in a stereo image pair are identified, the 3D depth can be easily calculated by triangulation to obtain the final depth map. There are many different algorithms for computing stereo matching and a large number of implementations of these algorithms [2–7]. However, binocular depth estimation requires complicated lens designs

and costly post-processing techniques to reconstruct the digital terrain model (DTM) from a captured image pair. In some cases, it is difficult to find the corresponding points in scenarios where little or no texture is available, and the computational time required to process stereo image pairs to create the DTM can be high. For this reason, depth estimation has been considered a difficult field for many years. With the advent of deep learning, researchers have proposed and applied various methods that provide interesting results for monocular depth estimation [8–11]. In this process, the main model finds an identity function to approximate the ground truth of the real DTM, more precisely to reflect it in a three-dimensional space using only a two-dimensional source. In [12], they proposed transforming the depth regression task into a classification task, dividing the depth range into a fixed number of bins of fixed width and achieving improvements in depth estimation. Again, in [13], the authors dynamically compute the bins depending on the input data and predict the final depth values as a linear combination of bin centers without discretizing the depth values due to the classification approach, as in [12], increasing the quality of the results. In the last eight years, researchers have proposed generative adversarial networks (GAN), which have achieved promising results in generating complex images guided by supervised ground truth signals. GANs use a competition technique that uses two parts (generator and discriminator) that fight against improving the prediction output. The generator can be thought of as the counterfeit model, which tries to fool the discriminator model (police) by reproducing a fake output very similar to the real ground truth [14]. In a super-resolution context, GANs can be applied successfully, because they can reproduce the fake output as much as possible according to the ground truth available. In recent years, sophisticated models based on GAN have been proposed. They use models and various optimizations to achieve an enhanced super-resolution output [8,15]. The logic is to downsample the high-resolution ground truth by $N\times$ (usually $4\times$) and force the model to give an output that is the same $N\times$ upsample. The generalization achieved by the model will reproduce super resolution by $N\times$ in the evaluation step without any downscaling. Recently, a sophisticated method has been introduced by [16], which uses a sub-network to manipulate the interpolated downsampled image (without supervision signal) and feed into GAN, with the goal of improving the fake prediction output. In [17], the authors introduced a selective feature fusion (GLPDepth) in with global and local attention mechanisms are used to improve the depth prediction. They use an encoder-decoder model with lateral connections (such as the UNet model), which is useful to feed into their modules and merge multi-scale local features, with the global decoding recovering better fine details. Although these models are efficient, and the authors have proven the effectiveness of their proposals, our motivation is to unify super resolution and monocular depth estimation into one end-to-end model. Thus, the following questions arise naturally:

> *Is it possible to increase the resolution of the images by $4\times$ and, at the same time, estimate a DTM, always by $4\times$, through a single end-to-end model?*

> *Does a refinement learned from a low-resolution interpolated source during the training allow us to obtain better features?*

> *Does using the GAN approach through the model proposal offer performance advantages over a classic generative architecture?*

We would emphasize that contrary to stereo matching, which uses a pair of images (right/left) to generate the DTM final product, the model proposed in this paper uses only a single HiRISE input grey-scale source to predict the final super-resolution images and associated DTM. The main novelty of the proposal is in using the $4\times$ super-resolution output grey-scale and DTM images to generate a 3D final map using a single end-to-end model and a sub-network grafted onto the main model, which is able to improve the interpolated input source in the training step. Monocular depth estimation has experienced great demand in recent years, due to the wide availability of a single camera in most real-world application scenarios and due to it covering a wide range of contexts, such as drone navigation, virtual reality, and autonomous driving. The paper is organized as

follows: In Section 2, we describe in detail the dataset built. In Section 3, the proposed neural model is able to predict DTM and grey-scale images in super resolution. In Section 4, the quantitative/qualitative results of all experiments reported, and finally, in Section 5, the conclusions are presented.

## 2. Data

In this section, we describe the details of the data type used in this research, the dataset creation step, and data normalization pre-processing.

### 2.1. Satellite Sources

The High-Resolution Imaging Science Experiment (HiRISE [18]) is a powerful camera onboard the NASA Mars Reconnaissance Orbiter, which was launched in 2005 and arrived on Mars after one year. The camera operates in visible wavelengths with a high-resolution capability of 0.25 m/pixel, producing images with a high detail level never seen before in planetary exploration missions. The huge amount of data provided allows an analysis of unprecedented views of science targets that is useful for helping to select landing future sites for rover and human exploration. The Orbiter's altitude can vary from 200 to 400 km above Mars, and it can acquire images containing up to 28 Gb (gigabits) of data every 6 s. In Section 2.2, we provide a full description of the dataset creation, using all sources available in the HiRISE repository [19].

### 2.2. The Dataset

The training dataset is composed of 430 DTMs and the associated surface grey-scale images. DTMs have a spatial resolution of 1 m/pixel, while grey-scale images can have a spatial resolution varying from 0.25 m/pixel to 2 m/pixel. We upscale through interpolation (nearest neighbour resampling) the grey-scale image from 0.25 m/pixel to the same spatial resolution of the DTM available (1 m/pixel), such that the prediction output size and the pixel per pixel correlation between both sources are correlated (image and DTM). We collected all sources through the University of Arizona's HiRISE site [19] (accessed on 1 March 2022), which is shown in Figure 1 for some samples. To perform the experiment, we split these 430 pairs into 360 for the training set and 70 for the testing set. To train the model, we extract tiles from the original DTM and image.

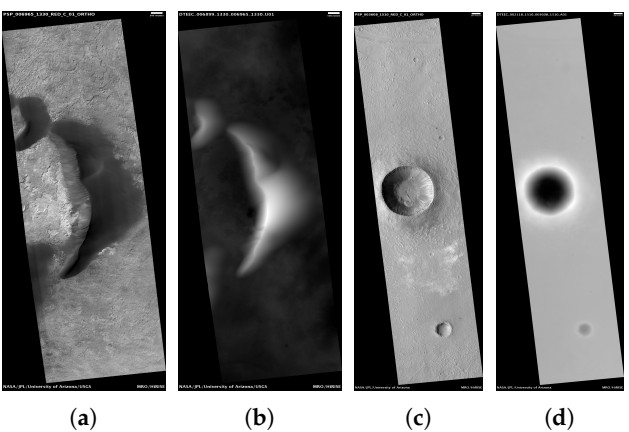

(a)  (b)  (c)  (d)

**Figure 1.** Overview of digital terrain models and their respective grey-scale sources for two random samples from the HiRISE repository. (**a**) Avalanche features of dune grey scale. (**b**) Avalanche features of dune DTM source. (**c**) Zumba crater grey scale. (**d**) Zumba crater DTM source.

Given that $h, w$ are the height/width of each tile extracted, we create the training set as follows:

$$S \times N \times h \times w$$

where the first dimension represents a DTM tile pair and its relative grey-scale image, and $N$ is the number of tiles extracted. More precisely, we generated $2 \times 120{,}000 \times 512 \times 512$ tiles from the training set and $2 \times 30{,}000 \times 512 \times 512$ from the test set. We discard some DTMs and the relative image correlated for anomaly height value (e.g., NaN value or absolute min value), and no data augmentation is applied. Finally, DTMs and grey-scale sources are scaled using max-min local normalization in a range of $[0, 1]$. We define the pixel range categorization as the total pixels that are in a specific range over the sum of all pixels of the test set. We report, in Table 1, the values in percentages.

**Table 1.** DTMs train/test set statistics using pixel-wise categorization based on the relative range. The total pixels of the HiRISE train and test set are respectively 31,548,243,968 and 7,883,194,368 pixels, with a depth values between $-7368.2$ and of 6871 m.

| Range (m) | Train Total Pixels | Percentage | Test Total Pixels | Percentage |
|---|---|---|---|---|
| $(-7500; -6750]$ | 125,167,881 | 0.40% | 36,128,083 | 0.46 % |
| $(-6750; -6000]$ | 402,230,537 | 1.27% | 98,323,693 | 1.25 % |
| $(-6000; -5250]$ | 212,877,177 | 0.67% | 53,962,715 | 0.68 % |
| $(-5250; -4500]$ | 1,638,429,900 | 5.19% | 380,020,058 | 4.82% |
| $(-4500; -3750]$ | 4,373,169,629 | 13.86% | 1,108,666,596 | 14.06% |
| $(-3750; -3000]$ | 4,249,001,383 | 13.47% | 1,056,827,070 | 13.41% |
| $(-3000; -2250]$ | 4,455,583,194 | 14.12% | 1,131,272,379 | 14.35% |
| $(-2250; -1500]$ | 4,086,800,593 | 12.95% | 1,014,990,398 | 12.88% |
| $(-1500; -750]$ | 2,268,149,838 | 7.19% | 571,919,007 | 7.25% |
| $(-750; 0]$ | 2,024,401,066 | 6.42% | 491,264,802 | 6.23% |
| $(0; 750]$ | 1,963,805,715 | 6.22% | 480,032,232 | 6.09% |
| $(750; 1500]$ | 2,664,539,777 | 8.45% | 674,413,972 | 8.56% |
| $(1500; 2250]$ | 1,683,998,656 | 5.34% | 426,284,350 | 5.41% |
| $(2250; 3000]$ | 504,330,832 | 1.60% | 136,542,305 | 1.73% |
| $(3000; 3750]$ | 452,791,812 | 1.44% | 113,674,919 | 1.44% |
| $(3750; 4500]$ | 274,299,407 | 0.87% | 70,333,088 | 0.89% |
| $(4500; 5250]$ | 7,744,087 | 0.02% | 2,035,558 | 0.03% |
| $(5250; 6000]$ | 32,818,301 | 0.10% | 6,395,581 | 0.08% |
| $(6000; 6750]$ | 72,529,655 | 0.23% | 18,573,226 | 0.24% |
| $(6750; 7500]$ | 55,574,528 | 0.18% | 11,534,336 | 0.15% |

*2.3. Data Normalization*

Normalization is a process used in data preprocessing and ensures a fast learning convergence if the data input $x \in [0, 1]$, because it removes the difference in magnitude between features data [20]. During the training step, all the learned weight model updates will have the same sign of the scalar error computed by the loss function and considering the input normalized and the linear/non-linear functions used by the model, this normalization process avoids the *zigzagging* weights behaviour, which can slow the learning step (Section 4.5 of [21]). In all experiments, a max/min local normalization is applied for each tile of the dataset in the loading data step of the training phase, taking full advantage of convergence according to the above-mentioned issues.

## 3. Methodology

In this section, we describe the proposed deep model, which is capable of estimating the DTM at a super-resolution scale and, at the same time, super resolution of the input grey-scale image. In Section 3.1, we introduce the details of the sub-network that is able to refine the interpolated input image, which is useful in improving the learning process, and in Section 3.2, we introduce the entire model summarized in Figure 2 and detailed in Figure 3.

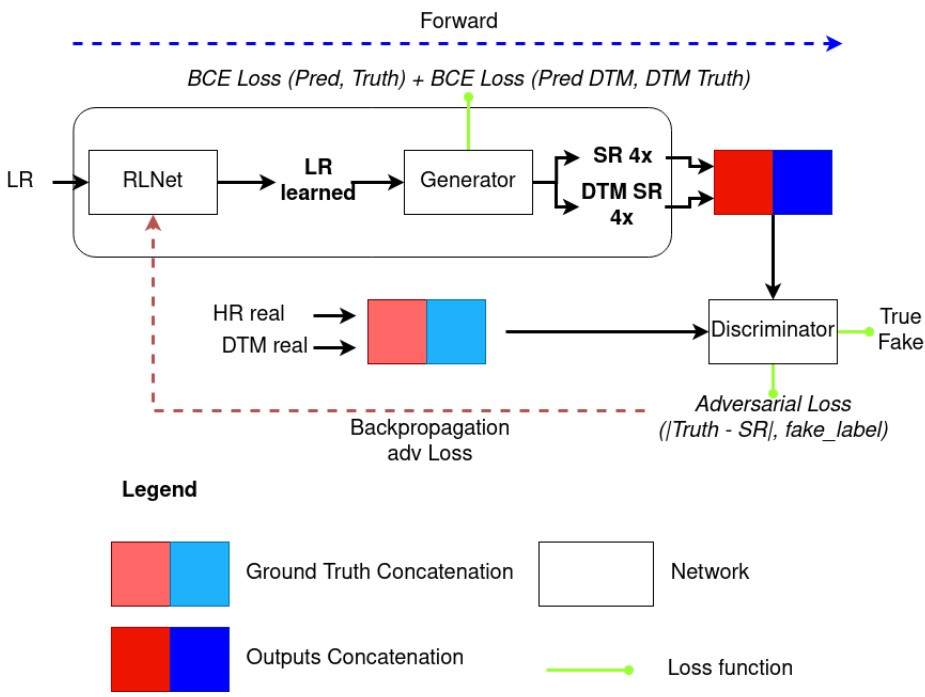

**Figure 2.** Training overview of SRDiNet + RLNet and generative adversarial approach (which we refer to as Model A).

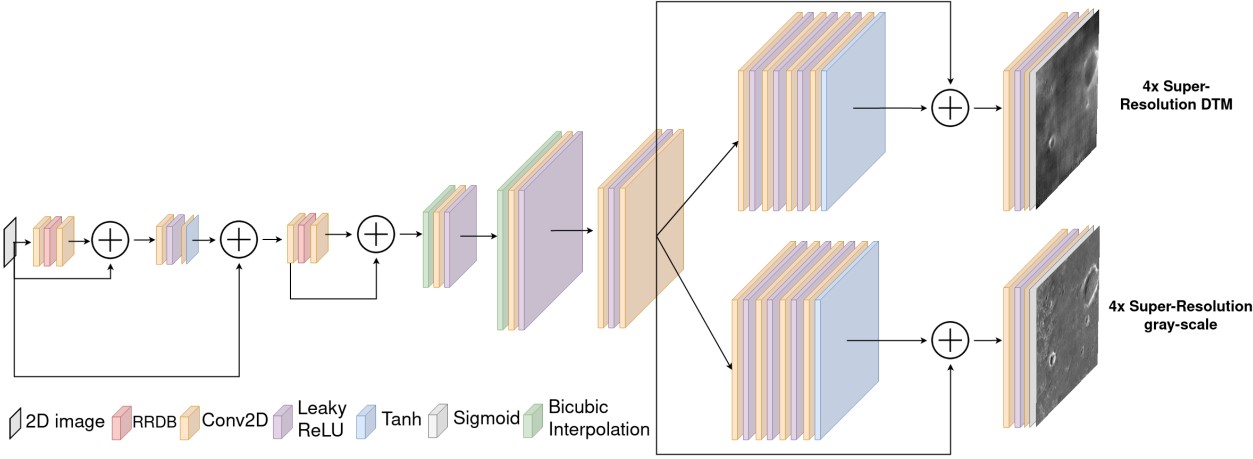

**Figure 3.** Architecture of SRDiNet jointly with RLNet sub-network.

### 3.1. Refinement Learned Network

The refinement learned network (RLNet) (see Figure 4) is responsible for refining the $I_n^{HR}$ interpolated in $I_n^{LR}$ outputs through a simple network with the final *Tanh* activation function that maps the features in a range $[-1, 1]$. We use the residual in residual dense blocks (RRDB blocks) [22] (short skip connection) to extract local features from the preceding blocks and preserve residual local information; then, the final features are summed up with the convolutional output of the low resolution, followed by a convolutive layer. The final features will sum over the image $I_n^{LR}$ through a long skip connection [23] to preserve the global features and induce a refinement effect over the input low-resolution image. It is demonstrated that short skip connections and long skip connections [23] are very useful to improve the quality of the features because providing alternative paths for the gradient computation of the backpropagation process is beneficial in the learning step. The proposed combination is trained using an unsupervised approach back-propagating the error of the last layers of the two-branch generator (Details in Section 3.2). Since the proposal is a unique end-to-end model, the refinement RLNet operation tries to regularize the $I_n^{LR}$ such

that the total error of the loss function is in a global minimum or a good approximation of it (local minima).

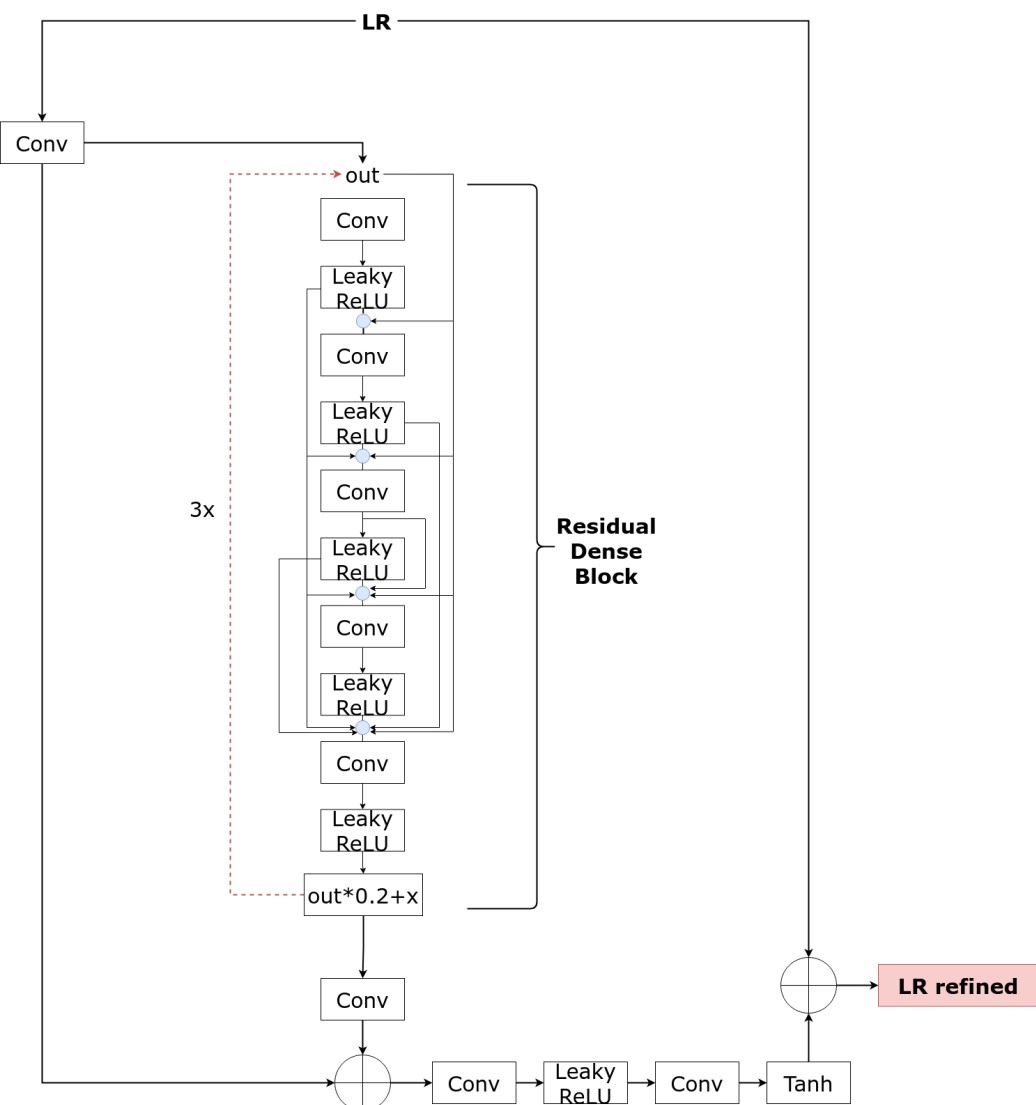

**Figure 4.** Refinement learned network (RLNet) architecture. The sub-network is grafted onto SRDiNet (Model A and Model B) to refine the interpolated input and improve the learning process.

### 3.2. Generator Two Branches Network

The SRDiNet model estimates the high resolution in an approximation super-resoluted image $I^{SR}$ from a low-resolution $I^{LR}$ through a learned weight from the scheme on Figure 4, such that the image feed into the generator network is $I^{RLR} = I^{LR} + \tanh(RLNet(I^{LR}))$. SRDiNet adopts a supervised approach, and all the high-resolution images and correlated DTM are provided as supervised signals. The SRDiNet model adopts a GAN-similar architecture using an RRDB block [22] to extract local features from the preceding and current blocks and stabilize the training of the wider network. Bicubic interpolation (upsampling layers with convolutive layers and leaky rectified linear unit activation function, referred to as Leaky ReLU) is used to enlarge the image, and then two-branch sub-networks are grafted to split the output along a unique end-to-end model (see scheme in Figure 3) and regularized by the coarse and fine local features fused jointly and fed into the last two sub-networks, which is useful to obtain better fine details. The sigmoid activation functions are used to scale the output data in a range of $[0, 1]$ as the input-normalized data. The final outputs will be the super resolution of the input source by $4\times$ and the relative estimated DTM map by $4\times$. The *Tanh* activation function maps the output of the local features learned

in a range of $[-1, 1]$, and summed with the feature data of the layer before the branch block, it has the scope to refine the features value of the *Conv4* output layer.

More formally, the generator process can be described as follows:

$$
\begin{aligned}
\alpha &= conv_1(I^{RLR} = I^{LR} + \tanh(RLNet(I^{LR}))) \\
out &= \alpha + conv_2(RDB(\alpha)) \\
out &= conv_6(conv_5(R(conv_4(R(conv_3(out)))))) \\
out_{dtm} &= out + \tanh(conv_{br0_{10}}(R(conv_{br0_9}(R(conv_{br0_8}(R(conv_{br0_7}(out)))))))) \\
out_{grey} &= out + \tanh(conv_{br1_{10}}(R(conv_{br1_9}(R(conv_{br1_8}(R(conv_{br1_7}(out)))))))) \\
pred_{grey} &= \sigma(conv(R(conv))) \\
pred_{dtm} &= \sigma(conv(R(conv)))
\end{aligned}
\tag{1}
$$

where $conv_n$ denotes the th-convolutional operation, $conv_{br}$ denotes the convolutional layer of the branch ($conv_{br0}$ and $conv_{br1}$), RDB denotes the residual dense block layer, and $pred_{dtm}$ and $pred_{grey}$ denote the estimated super-resolution DTM tensor and the super-resolution image of the input image $I^{LR}$. In addition, $\sigma$, tanh are the activation functions used, and Leaky ReLU $R(x)$ is defined as follows:

$$
R(z) = \begin{cases} z & z > 0 \\ \beta z & z <= 0 \end{cases}
\tag{2}
$$

For each training of the correspective triple sources $I_n^{HR}, I_n^{LR}, DTM_n^{I_n^{HR}}; n = 1, \ldots, N$, we search:

$$
\tilde{\theta}_G = arg\ min \left( BCE(G_{\theta_0}(RLNet(I_n^{LR})), I_n^{HR})) + BCE(G_{\theta_1}(RLNet(I_n^{LR})), DTM_n^{I_n^{HR}}) \right)
\tag{3}
$$

We defined the multi-objective loss function in Equation (3). The partial derivatives for each input-output pair can be expressed as follows:

$$
\frac{\partial E(X, \theta)}{\partial w_{ij}^k} = \frac{1}{N} \cdot \sum_{d=1}^{N} \frac{\partial}{\partial w_{ij}^k}(E) = \frac{1}{N} \cdot \sum_{d=1}^{N} \frac{\partial E_d}{\partial w_{ij}^k}
\tag{4}
$$

where the weight $w_{ij}^k$ connects the output of node $i$ in layer $k - 1$ to the input of node $j$ in layer $k$ in the computation graph, and $E$ is the binary cross-entropy (generalized in Equation (3)), more formally expressed as follows:

$$
E = -w_d \cdot [y_d \cdot \log(\tilde{y_d} + (1 - y_d) \cdot \log(1 - \tilde{y_d})]
$$

To generalize the main workflow and improve the readability, we highlight the entire proposal model as follows: Given a single grey-scale image, we feed it into SRDiNet, which predicts DTM and grey-scale sources in super-resolution thanks to the two-branch networks (see Figure 3). In Figure 2, an overview of the training workflow (Model A + GAN approach) is shown, with the forward and backward errors highlighted by blue and red lines, respectively. The discriminator network is used following the guidelines of the generative adversarial networks, which force the entire model to generate outputs very similar to the ground truth. The generator network back-propagates the error given by Equation (3) plus the adversarial loss function value (see Equation (5)) computed from the output of the discriminator (GAN approach). Furthermore, the discriminator is trained as a classifier to recognize true instances from among the fake ones generated. In the inference step, only the trained generator and RLNet are used to produce both super-resolution outputs. Finally, both generated sources are used in post-processing to build a 3D map (see Figure 5). To demonstrate the impact of SRDiNet and the effectiveness of the RLNet, we decide to create three variations of the above-mentioned proposal with/without

the GAN approach and with a detached/attached RLNet (See Table 2); we show a full overview (Model A) in Figure 2, and major details of the RLNet+Generator proposed in Figure 3. Using the GAN approach (Model A and Model C), which uses the discriminator network, we define $hr_{package}$ and $sr_{package}$ as concatenation along the channel axis between $(HR, DTM)$ and $(SR, DTM_{pred})$, respectively. Then, we compute and back-propagate the batch average error considering Equation (3) jointly with the following adversarial loss:

$$\mathcal{L}_{adv} = \lambda_{adv} \times \sum_{1}^{N} -\log(1 - |hr_{output} - sr_{output}|) \tag{5}$$

where $\lambda_{adv} = 1e^{-5}$, $hr_{output} = \text{discriminator}(hr_{package})$, and $sr_{output} = \text{discriminator}(sr_{package})$. The learning of the discriminator network is guided by the binary cross-entropy using the $sr_{package}$ flagged as *Fake* and $hr_{package}$ as *True*, accordingly with the main references [14,24]. In all experiments carried out, we use the ADAM optimizer [25] with a learning rate 0.0001 and cosine annealing [26] to decrease to a minimum value the learning rate over the epochs set up (50 epochs).

**Table 2.** Different variations of SRDiNet.

| Model Type | RLNet | Generator | Discriminator |
|---|---|---|---|
| Model A | ✓ | ✓ | ✓ |
| Model B | ✓ | ✓ | |
| Model C | | ✓ | ✓ |

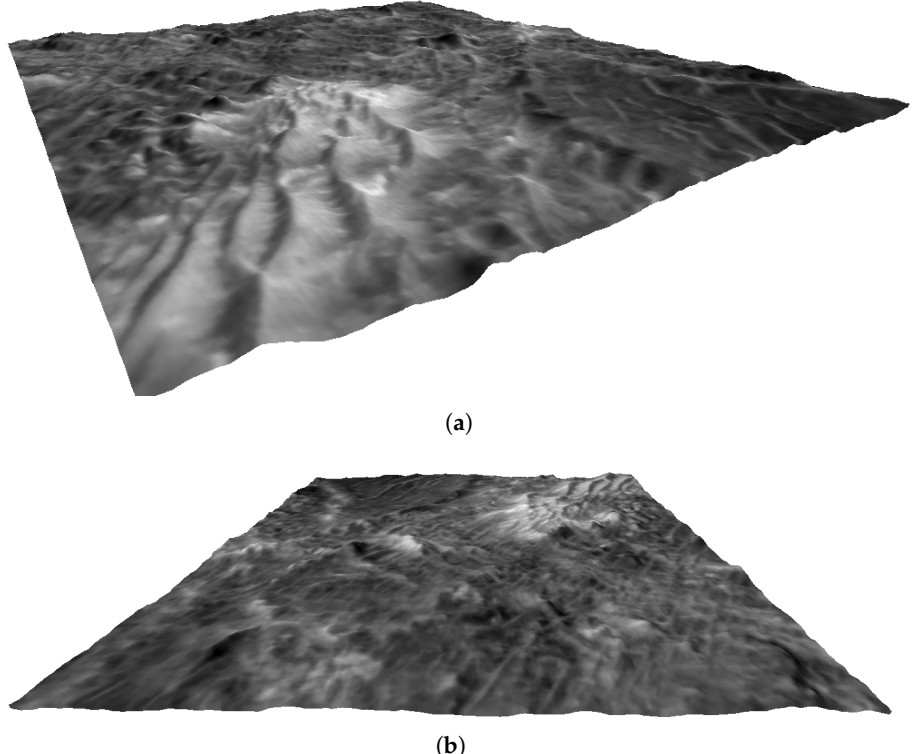

(a)

(b)

**Figure 5.** Two different 3D visualizations of the same region (**a**,**b**) using the outputs (DTM and grey-scale both in 4×) of SRDiNet starting from a single 512 × 512 tile of the HiRISE dataset with spatial resolution of 0.25 m/pixel and without DTM input.

## 4. Results

### 4.1. Quantitative Analysis

In order to study the effects of each model's variation of the proposed SRDiNet, we compare the differences using all the metrics reported in Equation (6), outlining the results over 30,000 instances of the HiRISE test set (see Table 3). Both Model A and B, which use RLNet, outperform Model C, along all metrics considered, demonstrating the effectiveness of RLNet to refine the interpolated input and improving the learning process. In Figure 6, we show in false color the variation of the input-learned image by the RLNet (third column) compared to the interpolated input source (second column), showing the capability to attenuate or accentuate some regions of the input image. Two types of evaluation metrics (error and accuracy) are adopted to report the quantitative results and compare the image and the digital elevation map predicted by the model analyzed. The first type (smaller is better) is the absolute error and the root means square error (RMSE), while the second type (bigger is better) estimates the accuracy using PSNR or using the threshold $\delta < 1.25^t$, where $t = 1, 2, 3$ and $d_p$, $\hat{d}_p$ are the ground truth and the predicted depth, respectively. More precisely, these can be expressed as follows:

$$PSNR = 10 \cdot \log_{10}\left(\frac{MAX_I^2}{MSE}\right)$$

$$Absolute\ Error = \frac{1}{T}\sum_{p}^{T}|\ d_p - \hat{d}_p\ |$$

$$RMSE = \sqrt{\frac{1}{T}\sum_{p}(d_p - \hat{d}_p)^2}$$

$$Accuracy\ threshold = \max\left(\frac{\hat{d}_p}{d_p}, \frac{d_p}{\hat{d}_p}\right) = \delta < thr$$

(6)

**Table 3.** Experimental results on the HiRISE test set (30,000 instances) using the three variation models (Model A, Model B, Model C). The s1, s2, s3 values represent the threshold accuracies as we described in Equation (6).

| Metric (avg) | Model A | Model B | Model C |
|---|---|---|---|
| PSNR *SR/HR* ↑ | 25.400 | **26.456** | 25.689 |
| PSNR *DTM SR/HR* ↑ | **15.069** | 14.930 | 14.813 |
| RMSE *SR/HR* ↓ | 0.0567 | **0.0514** | 0.0552 |
| RMSE *DTM SR/HR* ↓ | **0.1859** | 0.1876 | 0.1903 |
| Absolute err. *SR/HR* ↓ | 0.0417 | **0.0371** | 0.0404 |
| Absolute err. *DTM SR/HR* ↓ | **0.1558** | 0.1574 | 0.1594 |
| s1 *SR/HR* ↑ | 0.9098 | **0.9233** | 0.9149 |
| s2 *SR/HR* ↑ | 0.9834 | **0.9853** | 0.9841 |
| s3 *SR/HR* ↑ | 0.9947 | **0.9951** | 0.9948 |
| s1 *DTM SR/HR* ↑ | **0.3967** | 0.3882 | 0.3834 |
| s2 *DTM SR/HR* ↑ | **0.6731** | 0.6697 | 0.6628 |
| s3 *DTM SR/HR* ↑ | **0.8208** | 0.8204 | 0.8158 |

In Table 4, we report the best model in depth estimation (Model A) and the relative comparison with the GLPDepth model [17]. Although the SRDiNet makes estimates, the super-resolved DTM, and the image source by $4\times$ from a single image, we outperform GLPDepth model [17] performance, which estimates only the DTM map without a super-resolution task. In Table 5, we report the absolute error computed by the pixel range category. Model B outperforms Model A in all ranges in which we have a lower pixel percentage presence, while Model A offers better performance in ranges with a high pixel availability Table 1. Model B variation uses a generator without an adversarial approach, and considering the few data of the ranges where it performs better than Model A, it can

generalize better because it does not use a discriminator to force the generator to better reproduce the input source and estimate the depth map. In compairson, Model A better generalizes all pixels in the middle range due to the high percentage of data availability. The performance of Model C (SRDiNet + GAN and without RLNet) offers lower performance in all ranges categorized.

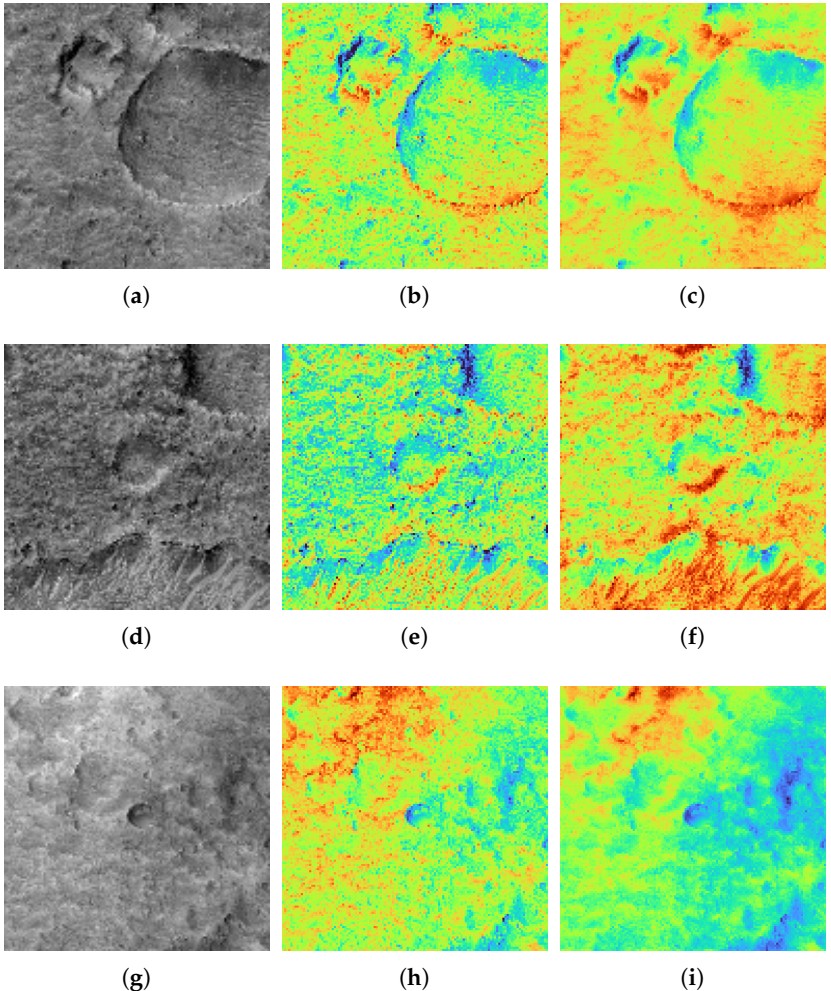

**Figure 6.** In (**a**,**d**,**g**) (first column), we show the original grey-scale sources after the classical bicubic interpolation; in (**b**,**e**,**h**), the relative bicubic sources in false colour (second column); and in the third column (**c**,**f**,**i**), we show the output of RLNet in false color. We highlight the pixel difference (blue low, red high) in the last two columns as a refinement over bicubic sources through the RLNet model to adjust some pixels so that the model improves the final prediction.

**Table 4.** DTM estimation of the entire test set of HiRISE (30,000 instances) and comparison between Model A and GLPDepth model [17] using all the metrics reported. Model A outperforms GLPDepth's performance over all results.

| Metric (avg) | Model A | GLPDepth [17] |
|---|---|---|
| PSNR *DTM SR/HR* ↑ | **15.069** | 14.5610 |
| RMSE *DTM SR/HR* ↓ | **0.1859** | 0.2310 |
| Absolute err. *DTM SR/HR* ↓ | **0.1558** | 0.1590 |
| s1 *DTM SR/HR* ↑ | **0.3967** | 0.3729 |
| s2 *DTM SR/HR* ↑ | **0.6731** | 0.5907 |
| s3 *DTM SR/HR* ↑ | **0.8208** | 0.7312 |

**Table 5.** Absolute error per range using three different model variations (Model A, Model B, Model C). The best results (lower is better) are in bold. We fix the ranges between max/min of the entire dataset with a step of 200 m for each of them.

| Range (m) | Model A | Model B | Model C |
|---|---|---|---|
| [−9000 −8200) | 0.1378 | **0.0621** | 0.1271 |
| [−8200 −7400) | 0.1531 | **0.0794** | 0.1499 |
| [−7400 −6600) | 0.1495 | **0.1012** | 0.1515 |
| [−6600 −5800) | 0.1501 | **0.1254** | 0.1524 |
| [−5800 −5000) | 0.1570 | **0.1434** | 0.1565 |
| [−5000 −4199.9) | 0.1607 | **0.1541** | 0.1609 |
| [−4199.9 −3399.9) | 0.1598 | **0.1593** | 0.1620 |
| [−3399.9 −2600) | **0.1588** | 0.1610 | 0.1621 |
| [−2600 −1800) | **0.1592** | 0.1619 | 0.1630 |
| [−1800 −1000) | **0.1605** | 0.1639 | 0.1652 |
| [−1000 −200) | **0.1635** | 0.1662 | 0.1685 |
| [−200 600) | **0.1672** | 0.1695 | 0.1715 |
| [600 1400) | **0.1706** | 0.1709 | 0.1745 |
| [1400 2200) | 0.1732 | **0.1673** | 0.1763 |
| [2200 3000) | 0.1749 | **0.1582** | 0.1775 |
| [3000 3800) | 0.1781 | **0.1437** | 0.1791 |
| [3800 4600) | 0.1817 | **0.1240** | 0.1817 |
| [4600 5400) | 0.1820 | **0.1015** | 0.1819 |
| [5400 6200) | 0.1609 | **0.0765** | 0.1785 |

*4.2. Science Case Study: Oxia Planum Site*

Characterizing the planetary surface, analyzing and investigating the geochemical process environment, and finding water clues represent necessary steps in the search for signs of past and present life on other planets. The ExoMars Rosalind Franklin rover is a sophisticated rover, and its landing on Mars will be expected in a few years. A strong landing site selection process (2014–2018) has been conducted by ESA, appointing the best one: the Oxia Planum region. The main reasons are due to the antiquity of the site (Noachian to early Hesperian, >3.6 Ga). Many studies suggest water-related activity (CITE) and have a particular interest in the presence of a clay-bearing unit, which might be a key target in the search for past biosignatures. More precisely, Oxia Planum is a 200 km-wide low-relief terrain characterized by hydrous clay-bearing bedrock units located at the southwest margin of Arabia Terra. This region exhibits Noachian-aged terrains. This location covers a key role in Mars exploration, and analyzing the super-resolution outputs in such regions ahead of the arrival of the rover can allow interesting observation and interpretation of the site's geology and future region selection. We select 1 m/pixel of spatial resolution over the HiRISE grey-scale dataset taking all Oxia Planum locations and dividing the huge maps into different $512 \times 512$ tiles. We highlight that all metrics reported are given by the downscale by $4\times$ the grey-scale images. We applied all metrics using the output predicted $4\times$ and the ground truth. The real super-resolution results are given in REF, in which we cannot report numerical results because the ground truth does not exist in that spatial resolution.

Real Super-Resolution Experiment

In this experiment, we select a full grey-scale image (0.25 m/pixel of spatial resolution) over the Oxia Planum Mars site [27], applying a "tiling" process to extract $7092 \times 512 \times 512$ unique locations. In Figure 7, we show some random locations between the GT and SR outputs.

We apply a filter color to highlight the different details (Figure 7a–c) captured by SRDiNet and the GT, demonstrating the capability of the model to capture fine image details and its generalization ability to enlarge the image from $512 \times 512$ to $2048 \times 2048$ and from a spatial resolution of 0.25 m/pixel to 0.06 m/pixel.

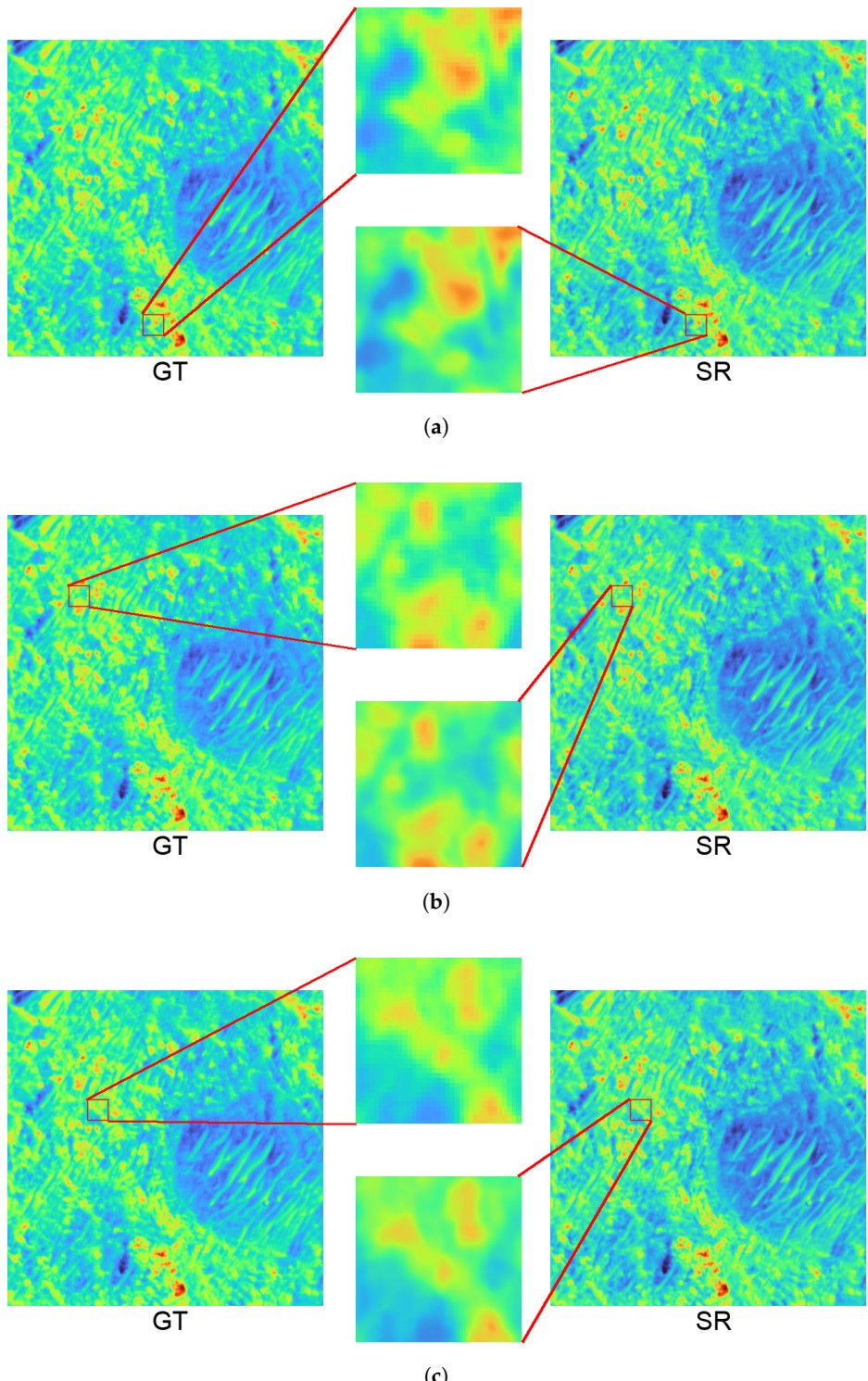

**Figure 7.** Random sub-locations of the full image *ESP 037070 1985* of the HiRISE dataset. The first column is the ground truth (GT), and the second is the output of SRDiNet by 4×, at 0.25 m/pixel of GT. The red squares contain the same locations of the tile analyzed. We suggest zooming in to observe fine difference details (**a**–**c**).

In Table 6, we compare SRDiNet with its variant using the above-mentioned metrics, highlighting better performance compared to the GLPDepth [17] depth estimation model. Once again, Model A (SRDiNet + RLNet + GAN approach) yields the estimated DTM

map better than its variants and in comparison with the GLPDepth model. In Figure 8, we analyze the DTM estimated, visualizing the ground truth (Figure 8a), the prediction (Figure 8b), and the absolute error between them (Figure 8c), showing a mean error value of 2.0275 meters per pixel (Figure 8d). In Figure 8e, the low-resolution GT and the super-resolution output of the SRDiNet are shown, demonstrating the generalization capability of SRDiNet to enhance the source in a super-resolution image and estimate the depth map, both by 4×.

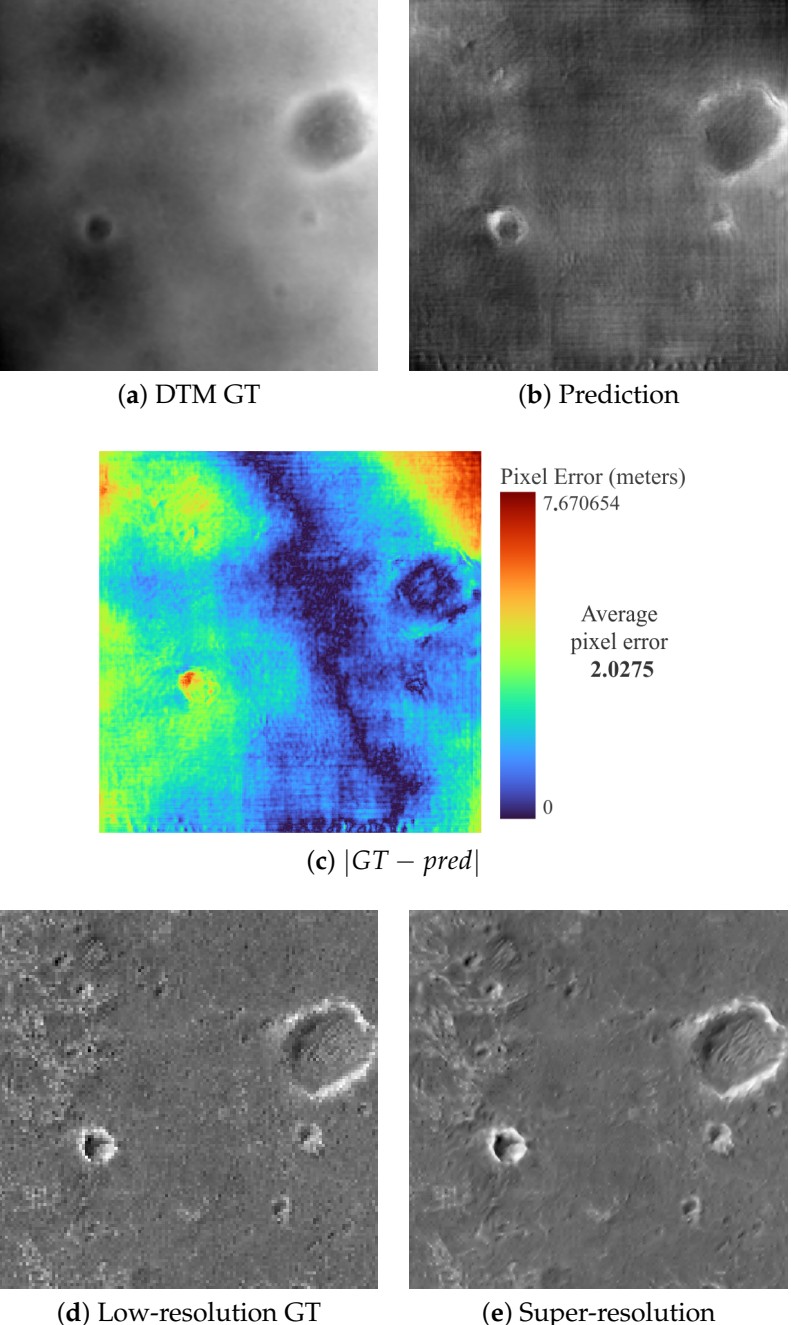

(**a**) DTM GT    (**b**) Prediction

(**c**) |*GT − pred*|

(**d**) Low-resolution GT    (**e**) Super-resolution

**Figure 8.** The GT (**a**) and DTM (**b**) estimation (super-resolution by 4×) from the low-resolution input target (**d**). In (**c**), the absolute error value between GT and DTM estimation is shown in false colors. We highlight a low average pixel error of 2.027 m. In (**e**), the super-resolution predicted image is shown.

**Table 6.** Comparison results of Oxia Planum site images through Model A, Model B, Model C, and a monocular depth estimation model known in the literature (GLPDepth model [17]). In bold, the best results reported show a better performance of model A compared to the others in DTM estimation.

| Metric (avg) | Model A | Model B | Model C | GLPDepth [17] |
|---|---|---|---|---|
| PSNR *DTM SR/HR* ↑ | **14.221** | 14.211 | 14.161 | 14.1184 |
| RMSE *DTM SR/HR* ↓ | **0.2011** | 0.2013 | 0.2026 | 0.2155 |
| Absolute err. *DTM SR/HR* ↓ | **0.1697** | 0.1700 | 0.1710 | 0.1787 |

In Figure 9, we track a terrain profile near a crater of the Oxia Planum site in order to analyze the pixels' similarity across a pixel path between the DTM ground truth (Figure 9a) and the SRDiNet output prediction (Figure 9b) using the corresponding low-resolution grey-scale image. The analysis of the terrain profile on the same path shows a high degree of similarity (see Figure 9c), with a low error reported in meters (y-axis). Although the values extracted by the pixels' path are very similar, we emphasize that the difference error showed in the y-axis is in a range of [0–1.5] error pixels, demonstrating the capability of the SRDiNet to approximate a hypothetical input ground truth. Finally, using the super-resolution estimated DTM and image sources, in Figure 5 shows a visualization of the 3D map exported by QGIS software of a 512 × 512 Oxia Planum tile super-resolved into 2048 × 2048 and from a spatial resolution by 0.25 m/pixel to 0.06 m/pixel. The input image fed into SRDiNet covers 128 m, and no DTM is provided in the test model. In Table 7, the computational inference time is reported using 4 × Nvidia RTX 5000 with 16,125 MiB of memory, each with a parallel approach to accelerate testing. Additionally, we are moving all instances for this test from external NAS storage to internal NVMe SSD storage with a maximum sequential read speed of 3500 MB/s, which is useful in improving the data loader function and decreasing the computational inference time.

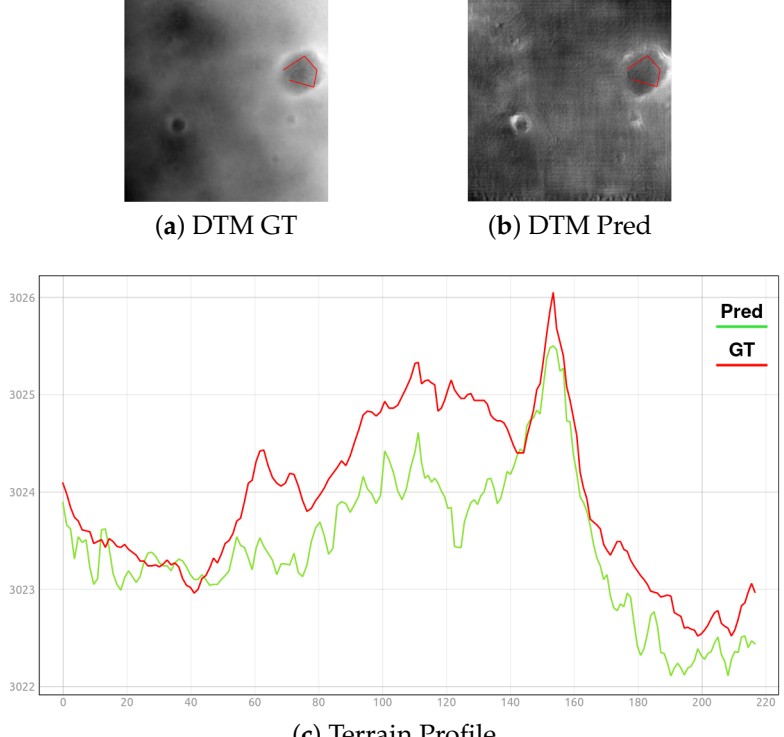

(**a**) DTM GT      (**b**) DTM Pred

(**c**) Terrain Profile

**Figure 9.** *Cont.*

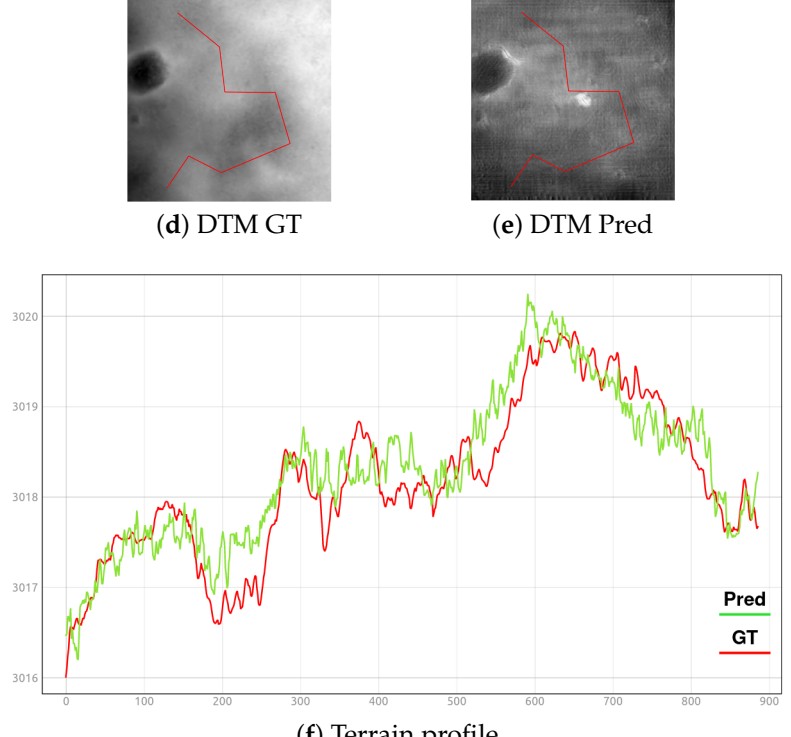

(**d**) DTM GT　　　　　　　(**e**) DTM Pred

(**f**) Terrain profile

**Figure 9.** Digital Elevation map: (**a**) GT, (**b**) DTM prediction. In (**c**), the same terrain profile tracked inside the crater shows a high similarity degree between the two DTM sources (**d**–**f**).

**Table 7.** Computational time (over HiRISE dataset) of the SRDiNet and its variations using the $512 \times 512$ tile size.

| Model | Parameters (M) | GPUs | Batch-Size | Instances | Inference Time (s) |
|-------|---------------|------|------------|-----------|--------------------|
| Model A/B | 15.35 | 4 | 4 | 500 | 246 |
| Model C | 12.39 | 4 | 4 | 500 | 218 |

### *4.3. Discussion*

In this section, a brief discussion is provided, highlighting the main results and the importance of the monocular depth estimation task in the space context. As shown in Figure 7, the advantages of super resolution applied to satellite data can help to understand and highlight details, and together with a reasonable estimate of the depth in super resolution, it is possible to obtain a 3D map of better quality (see Figure 5). The analysis of the terrain profile (DTM GT, DTM estimated) shown in Figure 9 has a high degree of similarity on all the pixels of the path traced in the figure; the same conclusion is reached also by analyzing the quantitative results through all the metrics considered, from which we can conclude that the proposed method is applicable in contexts similar to those analyzed in this paper. All quantitative results reported in Tables 4 and 6 demonstrate improvements compared to the GLPDepth model, and Model A overtook Model B and Model C in terms of DTM prediction. The main motivation to introduce three model variations is to prove the effectiveness of the generative adversarial concept inside model A and highlight the RLNet sub-network's ability to refine the interpolated source in order to achieve better results (Model A and Model B) compared to Model C (see Table 2 for details). The artifact generation is very difficult to produce using well-trained deep learning if the dataset used for the training step contains few artifact frequencies, and conversely, if the model has generated some artifact, it represents a hard challenge to detect. The main reason is the super-resolution mechanism, where the model will try to add fine details in all regions, and consequently, in regions with very low details or where they contain some error type, the main model will approximate substructures. A remarkable advantage of this proposal is to

obtain good DTM super-resolution products in a fast way, as the comparable commercial software is time-consuming, and for this reason, only a few samples have been generated (∼850 over 7100 stereo pairs). This approach can cover the remaining unavailable DTM products. Furthermore, the weighted learned features of the pretrained model can be used, and through a fine-tuning technique, it is possible to predict other targets in the spatial context. In Table 7, the benchmark reported shows the high usability in terms of computational inference time on 4 × GPUs. Although the hardware demand is high, we emphasize that the model enhances input data from a shape by 512 × 512 to a super-resolution output of 2048 × 2048 for both sources (DTM and grey-scale). However, it is possible to run the inference test in a single GPU with at least 12 G of memory, increasing the computational time and the training step using only batch-size 1.

A visualization tool is provided at the following link: https://huggingface.co/spaces/ARTeLab/DTM_Estimation_SRandD, and the Pytorch full code can be downloaded from [28].

In conclusion, the scientific contributions of this paper can be summarized as follows:

- Introduced a novel GAN model able to reproduce a super-resolution grey-scale image and predict DTM output by 4× feeding into the network only a grey-scale source.
- Built sub-network that refines the interpolated grey-scale image (unsupervised approach) and feeds into a model that uses the GAN paradigm, taking advantage of the predicted super-resolution outputs.
- Improvements in terms of architectural design finalizing into two-branch models able to predict both super-resolution outputs (DTM and grey-scale).
- HiRISE dataset creation useful to train all models built.
- Quantitative/qualitative analysis of three model variations on HiRISE dataset and comparison with a monocular depth-estimation model known in the literature.
- Analysis on a scientific case study over the Oxia Planum site (ESA-selected landing site for a future mission).

## 5. Conclusions

This paper proposed a novel architecture that can be used to enhance the resolution from a single 2D image to a 2D image by 4× and to generate a super-resolution DTM through a unique end-to-end model, referred to as SRDiNet. We carried out experiments using three variants of SRDiNet, describing the model details, dataset creation, loss functions, and training process. A model evaluation step using 30,000 instances is reported. and an interesting science site location study (Oxia Planum) was conducted, analyzing, visualizing, and reporting all the results, and demonstrating the effectiveness of the model to estimate both the high-resolution DTM and super-resolution 2D images. An advantage of this proposal is to obtain fast DTM super-resolution products, as the comparable commercial software is time consuming, and for this reason, only a few samples have been generated. The proposed model can cover the remaining unavailable DTM products, highlighting unseen details in all regions, even those with low spatial resolution in the monocular input image. Furthermore, we demonstrate improvements in generalization ability using the sub-network RLNet jointly with SRDiNet (Model A) compared to its variants and the GLPDepth model. In this paper, we successfully answer the question related to super resolution and depth estimation from a single input image that can be obtained using a single end-to-end model to create a 3D surface map in the planet science context.

**Author Contributions:** Conceptualization, R.L.G.; Data curation, C.R., G.C., N.L. and M.G.; Formal analysis, R.L.G.; Investigation, R.L.G. and I.G.; Methodology, R.L.G.; Resources, C.R. and G.C.; Software, R.L.G.; Validation, R.L.G.; Writing—original draft, R.L.G., I.G., G.C., C.P. and E.S.; Writing—review & editing, R.L.G. All authors have read and agreed to the published version of the manuscript.

**Funding:** This research received no external funding.

**Conflicts of Interest:** The authors declare no conflict of interest.

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
