# Peer review of "An Adversarial Generative Network Designed for High-Resolution Monocular Depth Estimation from 2D HiRISE Images of Mars"

_remotesensing, doi:10.3390/rs14184619_

Round 1
Reviewer 1 Report
Dear Authors,
Good effort has been made in the generation of DTM products. However, the manuscript needs to be formatted as per the requirements of the Journal with clear sections on Data, Methodology, Results, and Discussion; followed by a conclusion. English language / technical writing issues also need to be resolved. The methodology flowchart needs to be clear with a good follow-up understandable section/text in the manuscript. Currently, Figure 3 is quite dull, and it may be improved. Following are further suggestions for suitable manuscript improvement/corrections:
1. Line 5-8: Improper sentence, use tow simple sentences for: ”However, the entire reconstruction process 5 performed with classical stereo matching approaches can be time-consuming to compute and can 6 generate many artefacts which, coupled with the lack of adequate stereo coverage, can be a major 7 obstacle to 3D planetary mapping.”
2. Line19: Reframe “…We carried out studies and report the results for Oxia Planum” as “The results of Oxia Planum are reported….(Don’t use I/We/You etc. in manuscript). Write sentences as part of technical document.
- Line 66: Replace “which” by “with”
- Line 103: Replace “allows” by “allowed”
- Line 119/120: Clarify the combination of tiles. Currently its not clear (2 × 120, 000 × 512 × 512 tiles from the training 119 set and 2 × 30, 000 × 512 × 512×)
- Line 117/118/121/122/124/Caption of Figure 1/ 178 / 180 / 182/ 184/ 207/ 209/ 234/ 236/237/ 241 /244/ 251/252/ 254/ …./ 273 /281 : rephrase sentences with “we” without usage of i/We/you/our; as a practice for Journal papers/manuscripts.
- Line 126: Replace “ Tab. 1” as “Table 1”. Do same for all the tables.
- Rewrite section 2.3 clearly.
- Line 144: Provide full forms of all the terms throughout the manuscript on the first usage, for example: “RRDB block” / “ReLU” /”Leaky ReLU”?
- Line262: Correct/simplify the sentence(s): “…by not the same, we emphasize…”
- The Data section needs to focus on data with their details.
- The Methodology section shall be improved by focusing on the aim of clear objective(s).
- Results shall also focus on the intended procedure and output with intermediate results. The current writeup does not focus properly on the use of mono images for DTM/DEM generation while focusing on GAN.
- Explain all the parameters clearly while mentioning them in the procedure.
- Conclusion needs to be rewritten completely.
- Captions of the figure shall be made clear with the use of “we” in the sentences.
Best wishes,
Reviewer 2 Report
The paper proposes a fast and accurate methodology for monocular depth estimation from a super-resolution image exploiting a GAN-based model namely SRDiNet. Specifically, the developed network, pivoted on satellite imagery, generates a DTM prediction at 4x resolution as well as the super resolution prediction of the input gray-scale image. The learning is optimized by the refinement of the interpolated low-resolution input image performed by a sub-network. The evaluation tests and the comparison with existing model outline the SRDiNet’s efficiency in terms of generalization, depth map estimation and spatial approximation. However, it is not clear which cases/applications the generated super-resolution outputs can serve.
The methodology is presented with good clarity of style. The content is interesting and well-organized. Despite the fact that it is understandable by non-specialists, I would suggest to be more instructive and detailed regarding the description of SRDiNet and RLNet architecture (for instance the section 3.1). The topic is aligned with the aim and scope of the journal and the document can be considered a valid attempt toward the use of deep learning to monocular depth estimation.
The overall evaluation of the paper is positive. Nevertheless, some remarks can be referred to it and a minor revision in this sense would be welcome. Here is a list of points that should be reconsidered:
Introduction: The introduction is very detailed presenting the general area of interest, establishing the originality of the research and stating the current knowledge. A reference to the contributions of the work presented in this section and a short paragraph for the article’s structure description would be suggested.
Line 93: Consider to justify the applicability with some references.
Proposed Neural Model: Consider to make figure 1 and 2 more effective with a concise legend of the components being illustrated rather than a general description. You could also, briefly describe the essential components relevant to the figures.
Testing and Analysis: The quantitative metrics and benchmarking results may be further discussed to better support and extend the conclusions, such as for performance evaluation regarding CPU-GPU workload partition, memory footprint etc.
A discussion section is highly suggested in order to highlight the novelty and elaborate on the findings (strength and weakness, limitations, guidance to reproduce the experiments using the developed model, use cases etc.)
Round 2
Reviewer 1 Report
Dear Authors,
The introduction can be improved with the last para having the objectives/aim of the study. Contributions can come in Results and Discussion /Conclusion sections.
Some figures are not using the space properly, build in a very narrow area/layout. May be improved.
Figure 9, shall be made with the top (a), bottom (b), or as suitable with a clear caption.
Discussion - instead of a flat big paragraph, its content can be prepared in a combined section on "Results & Discussion", which will connect the right inferences/dis. at the right places along with figures/ graphs.
best wishes,
Author Response
Please see the attachment.
Thanks again for all your suggestions.
